# Comprehensive Ecotoxicity Studies on Quaternary Ammonium Salts Synthesized from Vitamin B_3_ Supported by QSAR Calculations

**DOI:** 10.3390/plants12040914

**Published:** 2023-02-17

**Authors:** Aleksandra Nowacka, Adriana Olejniczak, Witold Stachowiak, Michał Niemczak

**Affiliations:** Faculty of Chemical Technology, Poznan University of Technology, 60-965 Poznan, Poland

**Keywords:** nicotinamide, quaternary ammonium salts, white mustard, sorghum, plant development, *Lemna minor*, QSAR

## Abstract

Lately, ionic forms (namely, quaternary ammonium salts, QASs) of nicotinamide, widely known as vitamin B_3_, are gaining popularity in the sectors developing novel pharmaceuticals and agrochemicals. However, the direct influence of these unique QASs on the development of various terrestrial plants, as well as other organisms, remains unknown. Therefore, three compounds comprising short, medium, and long alkyl chains in *N*-alkylnicotinamide were selected for phytotoxicity analyses, which were conducted on representative dicotyledonous (white mustard) and monocotyledonous (sorghum) plants. The study allowed the determination of the impact of compounds on the germination capacity as well as on the development of roots and stems of the tested plants. Interestingly, independently of the length of the alkyl chain or plant species, all QASs were established as non-phytotoxic. In addition, QSAR simulations, performed using the EPI Suite™ program pack, allowed the determination of the products’ potential toxicity toward fish, green algae, and daphnids along with the susceptibility to biodegradation. The obtained nicotinamide derivative with the shortest chain (butyl) can be considered practically non-toxic according to GHS criteria, whereas salts with medium (decyl) and longest (hexadecyl) substituent were included in the ‘acute II’ toxicity class. These findings were supported by the results of the toxicity tests performed on the model aquatic plant *Lemna minor*. It should be stressed that all synthesized salts exhibit not only a lack of potential for bioaccumulation but also lower toxicity than their fully synthetic analogs.

## 1. Introduction

Vitamins are organic chemicals that play specific roles in the human body and are considered essential for maintaining optimal health. Vitamin B_3_, also known as vitamin PP or niacin, has three distinctive forms: nicotinic acid, niacinamide, and nicotinamide riboside [1]. These compounds are widely distributed in foods of plant and animal origin, however, their deficiency can lead to serious consequences such as pellagra, stomatitis, anxiety, fatigue, or depression [2,3,4]. Nicotinic acid can generally be found in plant products such as cereal bran or legumes, while nicotinamide is present primarily in animal products [5]. Both nicotinic acid and niacinamide are precursors in coenzyme biosynthesis: nicotinamide adenine dinucleotide (NAD^+^) and nicotinamide adenine dinucleotide phosphate (NADP^+^), which are necessary for the proper functioning of cell metabolism [6]. All three forms of vitamin B_3_ exhibit different effects on the human body, and, in general, the body handles nicotinamide better than niacin since it is more similar to NAD, resulting in greater bioavailability [7].

Since 1971, Lonza has been providing more than half of the world’s niacin needs in terms of human and animal nutrition through a well-designed organic synthesis pathway. One of the first methods used by the company is the oxidation of 5-ethyl-2-methylpyridine (MEP) with nitric acid [8,9]. In recent years, the company has also introduced technology for the production of nicotinamide by enzymatic hydrolysis of 3-cyanopyridine, using bacteria of the genus *Rhodococcus* [8]. Along with an increasing focus on the utilization of natural resources in organic synthesis, nicotinamide derivatives are gaining more and more interest, particularly in medicine and agrochemistry. An increasing number of studies have shown that many of the obtained derivatives exhibited anti-inflammatory, bactericidal, fungal, insecticidal, or herbicidal properties [10,11,12].

Many reports clearly indicate that quaternary ammonium salts (QASs), defined as compounds containing a positively charged nitrogen atom within the organic cation, are toxic to the majority of aquatic organisms, including fish, daphnids, algae, rotifers, and microorganisms employed in wastewater treatment systems [13,14]. Furthermore, some QASs have been proven to influence the development of higher plants and substantially inhibit their germination or growth [15,16,17,18]. Recent studies have shown that surface active QASs are also not inert to beetles and larvae that feed on plant matter [19]. All of these reports clearly illustrate the need to conduct research on the toxicity of QASs, including ionic derivatives of vitamin B_3_, which are currently lacking in the literature.

In accordance with the concepts of green chemistry and sustainable development, substances of natural origin are increasingly becoming an inspiration for scientists to obtain new chemicals that can exhibit a potentially lower impact on humans and the natural environment [17,20]. One such compound is vitamin B_3_, which in recent years has been successfully transformed into QASs, demonstrating many interesting biological properties and the potential for further intensive research. Therefore, in the framework of this study, three *N*-alkylnicotinamide bromides (presented in Figure 1) comprising different alkyl chains: butyl ([C_4_NA][Br]), decyl ([C_10_NA][Br]), and hexadecyl ([C_16_NA][Br]), were synthesized via quaternization of NA with an appropriate bromoalkane [21]. Subsequently, the obtained compounds were analyzed in the context of seeking a direct dependence between the length of the alkyl substituent and the assessed ecotoxicity. As a result, we describe the effect of the length of the alkyl substituent in the *N*-alkylnicotinamide cations, on the toxicity toward selected dicotyledonous (white mustard), monocotyledonous (sorghum), and aquatic freshwater (*Lemna minor*) plants. The species chosen for the study were selected on the basis of OECD guidelines [22].

It was assumed that the effect of NA modification towards QASs and the length of the alkyl substituent in the cation will be relevant in the context of the demonstrated ecotoxicity. Additionally, we hypothesize that the use of the vitamin as a building block will allow compounds to be obtained that are safer for the environment compared to fully synthetic QASs. Recently, ECOSAR, developed by the U.S. EPA, is being increasingly considered by many scientists as an accurate set of QSAR models that can reliably predict the toxicity of a given chemical to a variety of living organisms [23,24]. Therefore, ECOSAR was utilized to fill the data gaps for the ionic derivatives of NA in terms of their toxicity toward fish, daphnid and algae [25,26]. Furthermore, in order to reveal the products’ potential for bioaccumulation in the environment, their susceptibility to biodegradation was assessed according to well-known BIOWIN predictive models [27].

## 2. Results

### 2.1. Synthesis of Ionic Derivatives of Nicotinamide

The *N*-alkylnicotinamide bromides containing various alkyl chains responsible for different surface active properties [28]: weak-butyl chain in [C_4_NA][Br], medium–decyl chain in [C_10_NA][Br], and good–hexadecyl chain in [C_16_NA][Br] were synthesized according to the methodology described previously [21]. The obtained products were subsequently subjected to analyses allowing the determination of their physicochemical properties and ecotoxicity.

### 2.2. Water Solubility

In order to determine whether NA and its derivatives dissolve in water at a level that allows a toxic concentration to selected organisms to be achieved, their solubility in water was tested. The analysis was conducted according to the methodology given in Section 4.3. and calculated using the KOWWIN v1.68 software. The results are summarized in Table 1. The obtained experimental values were in the range of 0.024 to 873 g/L, and the predicted values occurred in the range of 0.0085 to 1000 g/L. In both applied methods the highest affinity for water exhibited the smallest ionic molecule-[C_4_NA][Br]. It was noted that the alkyl chain elongation contributed to a decrease in water solubility, hence, the lowest values were recorded for [C_16_NA][Br]. Interestingly, the solubility of NA determined by us experimentally (592 g/L) was similar to that present in the literature (500 g/L), while the predicted value was found to be nearly three times lower.

A discrepancy between both methods can easily be noticed, particularly for [C_10_NA][Br]. The tested values for this salt were nearly four times greater compared to the result calculated by the computer software. Among the tested compounds, the most hydrophilic salt [C_4_NA][Br] was the one that exhibited the lowest divergences between the experimental results and the predictions of KOWWIN v1.68.

The correlation between the results collected with the use of both methods is demonstrated in Figure 2. Ideally, all four points should coincide with the regressed diagonal line, which would imply a high accuracy between the measured and predicted values. In this case of NA and its derivatives, only one point ([C_4_NA][Br]) is fully consistent with the correlation line. The points for other compounds appeared slightly above the diagonal line, indicating that the variance in the error in the case of NA, [C_10_NA][Br], and [C_16_NA][Br] is constant across various levels of the dependent variable.

### 2.3. Octanol-Water Partition Coefficient

Bearing in mind that nicotinamide-based cations can be used as biologically active compounds, we investigated their partition between polar and nonpolar phases. These data allowed the determination of their potential environmental impact in the context of pollution of watercourses (for highly hydrophilic compounds) or bioaccumulation (for highly hydrophobic compounds). The logarithm of the octanol-water partition coefficient (log K_OW_) for NA, as well as its derivatives ([C_4_NA][Br], [C_10_NA][Br] and [C_16_NA][Br]), was examined experimentally (according to the methodology provided in Section 4.4) and using the WSKOWWIN v1.43 software. The results of this investigation are presented in Table 2. As can be seen, the measured log K_OW_ for analyzed compounds varied from −1.41 to 1.70, whereas the predicted values were much more diverse and ranged from −3.10 to 2.80. The lowest values were established for the ionic derivatives comprising the shortest alkyl chain ([C_4_NA][Br]) and, similarly in the case of water solubility, this parameter gradually increased with the alkyl elongation.

The least essential differences between the measured and predicted log Kow occurred for [C_10_NA][Br]. In this case, the values were almost identical and indicated the hydrophilic character of this salt (log K_OW_ greater than 0 suggests a hydrophobic character, while lower than 0 means that the compound is more hydrophilic). The least accurate predictions compared to the experimental values were demonstrated by [C_4_NA][Br]. In this case, the results differed by nearly 1.7 on the logarithmic scale, which means a difference of more than an order of magnitude in the concentration ratio. Interestingly, the log K_OW_ found in the literature for NA (−0.37) was closer to the result predicted in WSKOWWIN v1.43 (−0.45) compared to the value assessed experimentally (−0.66).

### 2.4. Ecotoxicity

#### 2.4.1. Toxicity toward Terrestrial Plants

The first phase of ecotoxicity studies focused on the determination of the toxicity of vitamin B_3_ and its ionic forms toward two model plants: white mustard (dicotyledonous plant) and sorghum (monocotyledonous plant). The research included the evaluation of the effect of selected compounds, applied in various concentrations, on germination as well as the early development of plants. All average stem and root lengths and the calculated standard error of the mean (SEM) are provided in Appendix A.

Analysis of the results obtained for white mustard leads to the conclusion that the tested solutions show neither toxic nor stimulating effects on the stems of the plants. Compared to the control, all other results for the white mustard stem are within tolerance levels. The lowest value developed by the stem amounted to 92%, whereas the highest value was equal to 109% of the control sample. Independently of the concentration of the tested compounds, the average results were found to be similar to those of the control. Statistical analysis revealed that there were no significant differences within all tested compounds, as well as all utilized concentrations (10, 100, and 1000 mg/kg, respectively). The determination of white mustard’s root development revealed that the compounds did not show statistically significant deviations in the length of the developed root compared to the control (range 72–106% of the control sample).

In the case of sorghum, the application of NA solutions at concentrations of 10 and 100 mg/L contributed to the mean stem lengths of 81% and 82% of the control, accordingly. At the same concentrations, [C_4_NA][Br] achieved similar results (80% and 69%), however, it should be noted that the differences were statistically insignificant. The results for [C_10_NA][Br] were the closest to the control sample in all the tested salts, clearly indicating the lack of any biological effect. It is noteworthy that the most surface active [C_16_NA][Br] demonstrated the greatest influence on sorghum stem development in the tested concentrations (85%, 71%, and 78% of the control), although the results were not statistically significant. On the contrary, none of the tested solutions showed signs of toxic effects towards sorghum roots (range 86–117% of the control).

#### 2.4.2. Toxicity toward Aquatic Live

To assess the aquatic toxicity category according to GHS classification, the toxicities toward fish, daphnids, algae, or water plants have to be assessed. The toxicity of vitamin B_3_ and its ionic analogs was predicted with the use of appropriately selected QSAR models (detailed explanation and other results are provided in Appendix A, and methodology was described in Section 4.5.3). As there were no QSAR models available to predict the toxicity of cationic surfactants toward algae or aquatic plants, range-finding tests on *L. minor* were performed experimentally to find the toxicity range of the examined compounds (detailed toxicities after 3 and 7 days are provided in Appendix A). As shown in Table 3, in the tests towards *L. minor* only compounds with decyl and hexadecyl alkyl constituents exhibited toxicity in the range of concentrations used, while the results of the other examined compounds indicate toxicity beyond the toxicity scale (above 1000 mg/L) [32]. [C_10_NA][Br] was practically harmless with toxicity in the range of 100–1000 mg/L, whereas [C_16_NA][Br] exhibited slight toxicity in the range from 10 mg/L to maximum water solubility of this compound (24 mg/L). All compounds did not show potential for bioaccumulation in organisms according to predicted bioconcentration factors [33] (detailed explanation in Appendix A).

NA and [C_4_NA][Br] were assessed as non-toxic to aquatic life, while [C_10_NA][Br] and [C_16_NA][Br] have been assigned to the ‘acute II’ category, according to GHS (Table 3). The elongation of the alkyl chain resulted in an increase in toxicity toward all considered organisms. The greatest differences were observed after the replacement of the butyl substituent for decyl. Eventually, salt with the longest chain (hexadecyl) was established to be the most harmful to aquatic species, with a predicted EC_50_ equal to 1.60 mg/L for daphnids and 1.90 mg/L for fish. Interestingly, the influence of NA and [C_4_NA][Br] on the functioning of terrestrial plants was noted as more significant compared to the predictions performed toward animals. By contrast, [C_10_NA][Br] and [C_16_NA][Br] exhibited an adverse trend.

### 2.5. Estimation of the Susceptibility to Biodegradation

The results from activated sludge models (Table 4) indicate that all compounds should biodegrade quickly (BIOWIN 1 & 2 > 0.5). However, the results of the MITI models (Japanese Ministry of International Trade and Industry) revealed contradictory conclusions (BIOWIN 5 & 6 < 0.5 means that all compounds are not readily biodegradable according to the OECD 301C test only). The compounds should undergo primary biodegradation within days (BIOWIN 4 close to 4.0), but ultimate biodegradation plausibly would require a few weeks (BIOWIN 3 close to 3.0) for [C_4_NA][Br], or weeks/months for the rest of the examined compounds. The results of BIOWIN 7 indicate that only NA can be considered readily biodegradable (BIOWIN 7 > 0.5).

### 2.6. Comparison of N-hexadecylnicotinamide Bromide with Its Ammonium and Pyridinium Analogs

The toxicity of one of the synthesized salts containing a hexadecyl chain within its structure ([C_16_NA][Br]) was compared with its ammonium and pyridinium analogs comprising the same length of the alkyl chain. The acquired data, presented in Table 5, indicate that hexadecyltrimethylammonium bromide [C_16_TMA][Br] and *N*-hexadecylpyridinium bromide [C_16_PIR][Br] can be considered as compounds exhibiting high hazards for aquatic life. In consequence, LC_50_ toward daphnids increased in the following order: 0.012 mg/L for [C_16_PIR][Br], 0.026 mg/L for [C_16_TMA][Br], and 1.60 mg/L for [C_16_NA][Br]. In the case of fish, the least toxic was [C_16_PIR][Br] (LC_50_ = approx. 40 mg/L). Due to the lack of data and suitable QSAR models, it was impossible to reliably compare the toxicity of these compounds towards *L. minor* or green algae.

Nonetheless, the investigated [C_16_NA][Br] turned out to be potentially the safest for the environment (‘acute II’ according to the GHS short-term toxicity criteria) compared to that of the other two fully synthetic analogs ([C_16_TMA][Br] and [C_16_PIR][Br]). Being in the ‘acute II’ class, the vitamin B_3_ derivative exhibits the most favorable LC_50_ values for fish and daphnids. Moreover, the data in Table 5 also indicate that all the analyzed substances do not show potential for bioaccumulation.

## 3. Discussion

One of the objectives of our research was to determine whether the tested solutions of NA and its derivatives affect the early development of seeds of model terrestrial plants (white mustard and sugar sorghum). Due to their plausible use as alternatives to currently used pharmaceuticals or agrochemicals, the focus on their effects on plants is extremely important. In recent years, much attention has been paid to the possible replacement of conventional agrochemicals with substances of natural origin due to potential economic and environmental benefits [21].

For the purpose of ecotoxicity studies on vitamin B_3_ and its derivatives using QSAR calculations, their selected properties were determined. The research involved an analysis of the solubility of the tested compounds in water as well as the determination of the octanol-water partition coefficient. Ecotoxicity studies were performed on terrestrial and aquatic model plants, while estimation of toxicity toward aquatic organisms (fish, daphnid, and green algae) and susceptibility to biodegradation was determined using EPI Suite programs.

According to the acquired data, due to the insertion of an ionic bond in the structure of the investigated vitamin B_3_ in the alkylation process, the products’ water solubility exceeded that of the starting compound (NA). However, it should be noted that the elongation of the alkyl chain attached to the nitrogen atom in the aromatic ring resulted in a decrease in their affinity for water. This dependence is in agreement with the values of the octanol-water partition coefficient, where the greatest affinity for water, characterized by negative log K_OW_ values, is exhibited by salt [C_4_NA][Br]. An increase in this parameter (and thus hydrophobicity) can be observed along with an increase in the number of carbon atoms in the alkyl chain, as the size of the nonpolar part of the molecule increases. Additionally, the range of log K_OW_ allowing the compound to be classified as ‘safe for the environment’ is between 0–3. Taking into account this fact, one of the tested compounds ([C_16_NA][Br]) can be defined as potentially eco-friendly [15,21,37].

In our previous study, *N*-alkylnicotinamide cations were combined with anions that are widely known as effective, selective herbicides, such as 2,4-D or MCPA. Interestingly, the synthesized new QASs were found to exhibit an enhanced herbicidal effect compared to the original herbicides [21]. However, a direct influence of the utilized cation on the development of the tested plants has as yet not been assessed. In effect, it is still unknown whether the cation is an inactive enhancer (adjuvant) or an additional phytotoxic agent that acts synergistically. In the case of the data collected in this study, the Kruskal–Wallis test was performed to determine if there is a statistically significant effect of the cation on the growth of stems or roots of white mustard and sorghum at various concentrations. Interestingly, the results of the study did not reveal statistically significant differences. In effect, independently of the alkyl length in the cation, as well as the concentration, the solutions did not show toxicity toward model terrestrial plants. Hence, the increase in the hydrophobicity of vitamin B_3_ derivatives in the tested range does not affect their phytotoxicity. Interestingly, according to research conducted by Biczak et al. in 2015, it was stated that, in the case of menthol derivatives, the toxicity decreases with increasing alkyl chain length (for alkyl chains containing 1–11 carbon atoms, the toxicity decreased 6-fold with alkyl elongation) [17]. Nonetheless, as a continuation of previous research [21], we found that nicotinamide-based cations most likely play the role of non-phytotoxic adjuvants, only enhancing the biological effect exhibited by the other substance (active ingredient). This is a particularly important discovery from the point of view of recent reports describing the high toxicity and carcinogenicity of commercially used adjuvants and opens up potential perspectives for the future development of agrochemicals based on NA [38].

Turek et al. in 2017, conducted research on other QASs—ammonium haloacetates—and analyzed their phytotoxicity toward spring barley and common radish to seek an alternative to glyphosate [39]. Studies have shown that the toxicity of the tested solutions in the case of both plants increases with the number of halo-substituents. Consequently, trisubstituted compounds were more toxic than disubstituted compounds. The toxicity of the tested compounds was also noted to be equally high for mono- and di-cotyledonous plants [39]. However, alteration of the alkyl substituent in the compounds’ cations did not contribute to an increase in their toxicity. Therefore, it was concluded that the toxic effect expressed by QASs is complex and can depend on various factors. In consequence, it is potentially possible to control the toxic effect by appropriate design of both cation and anion constituting QASs.

In 2016, Hong et al. conducted research on four vitamins, including vitamin B_3_, in terms of their antimicrobial activity against *Ralstonia solanacearum*, a pathological bacterium that infects more than 200 plant species. The obtained results indicated that vitamin B_3_ and other tested vitamins applied to tomato plants can reduce the bacterial wilt of plants, while not causing phytotoxic effects [40]. The results obtained by Hong et al. lead to the conclusion that vitamin B_3_ derivatives could potentially also be used as agrochemicals to protect crops against fungal infections, without showing a toxic effect on plants. A summary of the above-mentioned studies, investigating the toxic effect on terrestrial plants, is presented in Table 6.

In the case of derivatives of vitamin B_3_, the toxicity toward *L. minor* grows with alkyl chain elongation, starting from values above the toxicity scale (namely, above 1000 mg/L) [32]. In addition to an increase in toxic effect, the synthesized compounds exhibit a simultaneous enhancement in surface active properties due to the alkyl chain elongation [28]. Therefore, the toxicity originates from the presence of a long alkyl chain in the organic cation, which is well correlated with the available reports revealing that cationic surfactants can cause a disruption of the cell membrane [41]. However, the observed differences in toxicity between terrestrial and aquatic plants may be related to the fact that QASs are sorbed in the soil [42] and, as a result, are not absorbed by plants to the same extent as in the case of aquatic plants. The predicted toxicities (Table 3) indicate that compounds deprived of surface activity (NA and [C_4_NA][Br]) should be non-toxic, while salts with longer alkyls can be included in the ‘acute II’ category according to GHS. It should also be emphasized that the data in Table 5 indicate that synthesized salts exhibit a lower threat to aquatic life compared to other popular cationic surfactants, such as *N*-hexadecyltrimethylammonium bromide [C_16_TMA][Br] or *N*-hexadecylpyridinium bromide [C_16_PIR][Br].

According to the literature [34,35,36,43,44,45], *N*-alkylpyridinium halides are less toxic to fish and green algae, however, they exhibit similar toxicity to crustaceans to tetraalkylammonium analogs comprising the same length of the alkyl chain. Interestingly, derivatives of *N*-alkylpyridinium chlorides with ester bonds in the alkyl chain (so-called esterquats) exhibited 50-fold lower toxicity to zebrafish than their analogs deprived of a cleavable bond within the alkyl chain [46].

The association between the chemical structure and biological activity (SAR) of the QASs is an extremely complex problem that has been extensively studied in recent years, although no direct correlations have so far been discovered. QASs can act through various toxicity mechanisms, some of them demonstrate practically a lack of toxicity, whereas others can be extremely toxic (EC_50_ between 0.01 and 0.1 mg/L) to various biological systems [13,47,48]. Additionally, a literature survey reveals that the majority of amphiphilic QASs exhibit the potential to influence the functioning of various living organisms to a greater extent compared to their precursors [47]. However, according to the literature, the toxicity of QASs follows trends that are similar to ionic liquids (ILs) and depends on: (1) the length of the alkyl side chain in the cation; (2) the presence and the nature of functional groups present in the cation; (3) the nature of the anion and cation; and (4) interactions between the anion and cation [47,49].

The estimation of the susceptibility of vitamin B_3_ analogs to biodegradation provided predictions of high significance regarding their potential environmental impact. Although the results from activated sludge models (Table 4) indicate that all compounds biodegrade quickly, the results of BIOWIN 5 and 6 led to the conclusion that they are not readily biodegradable. It should also be mentioned that BIOWIN 7 implies that all the analyzed substances, except NA, do not decompose rapidly under the influence of biotic factors. However, according to ECHA, nicotinamide not only can be considered readily biodegradable (OECD 301F, 99% after 7 days) [50], but also some of its methylated derivatives (*N*-methyl-3-carbamoylpyridinium iodide and hexafluorophosphate) [51]. Therefore, further practical experiments are crucial in this field, which in the future will allow existing BIOWIN models to develop further.

Interestingly, the susceptibility to biodegradation of the previously examined pyridine derivatives exhibited a dependence on alkyl chain length, for example, *N*-ethylpyridinium, *N*-propylpyridinium, *N*-butylpyridinium, *N*-butyl-3-methylpyridinium halides did not exhibit significant levels of biodegradation and therefore were not metabolized by microorganisms [47,52,53]. *N*-hexyl-3-methylpyridinium bromide was established to be not readily biodegradable but biodegraded over an extended period of time, while *N*-hexadecylpyridinium, *N*-octyl-3-methylpyridinium, and *N*-(2-hydroxy)ethylpyridinium bromides met the criteria for ready and complete biodegradability according to the OECD test guidelines [47,52,53,54,55]. Similarly, pyridinium derivatives with carbonyl group connected to aromatic carbon or nitrogen atom such as 3-(butoxycarbonyl)-1-methylpyridinium and *N*-methyl-3-carbamoylpyridinium iodides or 1-(2-ethoxy-2-oxoethyl)pyridinium bromide were also assessed as readily biodegradable [51,56]. The literature analysis suggests also that QASs containing long alkyl side chains are more readily biodegradable despite possessing higher antimicrobial activity. Generally, the presence of sites for enzymatic hydrolysis (ester or amide bonds), and aromatic rings promote biodegradability [47]. The importance of the results gathered for the ionic derivatives of pyridine underlines the fact that the majority of conventional, fully synthetic QASs containing imidazolium, tetraalkylammonium, morpholinium, piperidinium, and pyrrolidinium cations are mostly resistant to biodegradation [47].

## 4. Materials and Methods

### 4.1. Materials

1-Bromobutane (99%), 1-bromodecane (98%), 1-bromohexadecane (97%), and 3-pyridinecarboxamide (nicotinamide, 98%) were purchased from Sigma-Aldrich (Saint Louis, MI, USA). The inorganic salts and all solvents were purchased from Avantor (Gliwice, Poland) and used without further purification. Deionized water with a conductivity < 0.1 μS cm^−1^, from the HLP Smart 1000 demineralizer (Hydrolab, Poland), was used. Phytotoxkits containing white mustard and sugar sorghum seeds as well as plastic containers and black filter contrast papers were purchased from MicroBioTests Inc. (Gent, Belgium). For the experiment, soil with the following elemental composition was used: 67 mg P kg^−1^, 55 mg K kg^−1^, 54 mg Mg kg^−1^, 100 mg Fe kg^−1^, pH 5.75 (in CaCl_2_), and a C organic content of 1.90% (19.00 g kg^−1^). *L. minor* plants were purchased from Perfekt Klik Sylwia Grzelak (Ostrzeszow, Poland) and the plants received an EU plant passport (A. Lemna B. PL-30/18/20385 C.3/20 D.PL).

### 4.2. Synthesis of N-alkylnicotinamide Bromides

Nicotinamide (NA) occurs naturally in many natural sources, however, for the purpose of this research, synthetic NA was used. Three ionic derivatives of vitamin B_3_*-N*-alkylnicotinamide bromides ([C_4_NA][Br], [C_10_NA][Br] and [C_16_NA][Br]), were synthesized via the quaternization reaction of nicotinamide with an appropriate bromoalkane in propan-1-ol at elevated temperatures. The precipitated products were isolated and purified according to the previously described protocol [21]. Ref. [21] contains spectral analysis and physicochemical characterization of [C_10_NA][Br] and [C_16_NA][Br]. The salt with the shortest alkyl chain [C_4_NA][Br] was previously synthesized and characterized by K. Kalyanasundaram et al. [28].

### 4.3. Water Solubility

The exact solubility in water was evaluated according to the OECD 105 guideline. Firstly, 0.1 g of the tested substance was placed in a vial and mixed with deionized water for 24 h, 48 h, and 72 h. The samples were then centrifuged and the liquid phase was collected using a syringe. Compound concentrations in water were determined spectrophotometrically using a UV/Vis spectrophotometer (based on previously made calibration curves with plots (at λ_max_ = 261 nm for NA, 264 nm for the other compounds) vs. concentration for each substance). Three replicates of each measurement were performed.

### 4.4. Octanol-Water Partition Coefficient

The log Kow was evaluated according to the OECD 107 guideline. K_OW_ values were assessed using mutually saturated distilled water and *n*-octanol in a glass vial with a magnetic stir bar. First, the NA and synthesized products were dissolved in distilled water, or octanol in the case of [C_16_NA][Br], in amounts corresponding to 0.01 M concentration. In the next step, stock water solutions were poured into vials with *n*-octanol in a 2:1, 1:1, or 1:2 ratio. All ratios were duplicated. All vials were shaken at 25 °C and after 24 h samples were centrifuged and then the aqueous and octanolic phases were collected by a syringe. The concentrations of compounds in water and in *n*-octanol were determined spectrophotometrically using a UV/Vis spectrophotometer (based on previously made calibration curves with absorbance plots in water and in *n*-octanol (at λ_max_ = 261 nm for NA, 264 nm for the analyzed salts) vs. concentration for each substance). Three repetitions of each measurement were performed.

### 4.5. Ecotoxicity

The ecotoxicity was tested in three stages. In the first stage of the research, the phytotoxicity of the tested compounds toward two model terrestrial plants was determined, while in the second stage of the research, the ecotoxicity towards the model water plant was determined. In the third stage, toxicity toward three representative aquatic species was predicted using appropriate QSAR models.

#### 4.5.1. Toxicity toward Terrestrial Plants

The experiment involved the assessment of the effect of synthesized salts on the germination and early development of model terrestrial mono- and di-cotyledonous plants. White mustard (*Sinapis alba* L.) and sugar sorghum (*Sorghum saccharatum* L. Moench) were tested using the phytotoxicity test based on the international standard ISO-11269-2:2003. The germination test was carried out in vertical plastic containers (Phytotoxkit, Tigret, Belgium). The containers were filled with 60 g of soil and then water solutions of the analyzed compounds (30 cm^3^) were added at a concentration of 10, 100, and 1000 mg kg^−1^. In the next step, 10 seeds of white mustard and sorghum were placed separately on the soil layer in a single row. The Phytotoxkit plastic containers were then closed and placed in the dark and kept at a temperature of 25 ± 1 °C for 7 days. As a reference sample, seeds were seeded in soil soaked only in distilled water. After the end of the experiment, the number of germinated seeds was counted, and the length of the roots and shoots of each plant was measured. For white mustard, after the removal and processing of the results, a second repetition was established in order to check the repeatability of the results. Comparable results for the second repetition confirmed the correctness of the research methodology and the absence of the need for additional tests for sorghum.

For all results, the standard error of the mean (SEM) was calculated according to the following equation:(1)$$\mathrm{SEM}=\frac{s}{n^{0.5}}$$
where SEM is the standard error of the mean, s is the sample standard deviation, and n is the number of samples.

#### 4.5.2. Toxicity toward *L. minor*

The plants were grown in covered plastic containers in modified pH-stabilized Steinberg medium [25,49,57] (3.46 mM KNO3, 1.25 mM Ca(NO3)2, 0.66 mM KH2PO4, 0.072 mM K2HPO4, 0.41 mM MgSO4, 1.94 µM H3BO3, 0.63 µM ZnSO4, 0.18 µM Na2MoO4, 0.91 µM MnCl2, 2.81 µM FeCl3, 4.03 µM EDTA; pH 5.5 ± 0.5) in a thermostated chamber at 25 ±2 °C. The chamber was continuously illuminated with an LED lamp with intensity in the range of 7000–10,000 lux. For all tests, the growth rate µ was used as the endpoint which is calculated on the basis of the number of fronds as:(2)$$\mu =\frac{\ln\left(F_{t2}\right)-\ln\left(F_{t1}\right)}{t_{2}-t_{1}}$$
where F_t1_ and F_t2_ are the numbers of the fronds on day t_1_ and day t_2_ of the experiment, respectively. From a mathematical perspective, µ is based on the assumption of exponential growth and gives an average of growth during the time period from t_1_ to t_2_. The growth rate inhibition was then calculated in relation to the control mean:(3)$$\%\mathrm{inhibition}=100\times \left(1-\frac{\mu_{sample}}{\mu_{control}}\right)$$

All tests were started with a minimum of 12 fronds per treatment. The number of fronds was determined at the beginning of the experiment and after 3 and 7 days. Compounds were tested at five concentrations (0.1, 1, 10, 100, 1000 mg/L) to find the toxicity range without renewal (static) of the test solutions. Each concentration was tested in five replicates.

#### 4.5.3. Predicted Toxicity towards Other Organisms

In order to predict the toxicity, the Ecological Structure Activity Relationships (ECOSAR) Class Program of the U.S. EPA (United States Environmental Protection Agency) was used. The program estimates a chemical’s acute (short-term) toxicity and chronic (long-term or delayed) toxicity by using computerized Structure Activity Relationships (SARs). Despite the presence of the ECOSAR 1.11 program in EPI Suite 4.11, ECOSAR 2.2 was used to ensure the best estimation by using the latest available software. For example, in the current version of ECOSAR (version 2.2), there are four general groups of surfactants. The previous ECOSAR version 1.11 included subgroups of surface-active compounds for ease of use only; the subgroups did not have separate QSARs. The SMILES used to enter the input data into the program were provided in Appendix A.

### 4.6. Estimation of Physicochemical Properties and Biodegradation with the BIOWIN Models

The water solubility, logarithm of the octanol-water partition coefficient (Log K_OW_), bioconcentration factor (BCF), and biodegradation of compounds used in this study were evaluated using programs implemented in EPI (Estimation Programs Interface) Suite 4.11. EPI Suite™ is a Windows ^®^-based set of physical/chemical property and environmental fate estimation programs developed by the U.S. EPA. Water solubility was predicted using KOWWIN v1.68, log K_OW_ was predicted using WSKOWWIN v1.43, BCF was predicted using BCFBAF v3.01, BIOWIN was predicted using BIOWIN v4.11. BIOWIN models predict the probability of biodegradation and duration of a substance in water and soil. BIOWIN is based on the group contribution approach in which a molecule is decomposed into its fragments, and biodegradability is estimated considering each fragment [58]. All BIOWIN models report numerical values, which are indicative of biodegradation. BIOWIN models 1–7 deal with the aerobic biodegradation of a substance in soil and water [27,59]. The SMILES used to enter the input data were provided in ESI in Appendix A.

## 5. Conclusions

Recent advances in the field of the derivatization of naturally occurring compounds have provided useful information on the proper adjustment of their properties, which opens up novel perspectives for their successful application. In pursuit of this strategy, a moiety of nicotinamide, commonly known as vitamin B_3_, has been transformed into quaternary ammonium salts (QASs) comprising different lengths of the alkyl chain attached to the pyridine ring. The performed study clearly confirmed that, independently of the length of the alkyl chain, the compounds did not affect the development of model terrestrial and aquatic plants (sorghum, white mustard, and *L. minor*). Due to the fact that the tested ionic derivatives of vitamin B_3_ show a lack of statistically significant differences in phytotoxicity compared to the control, they can be considered as potential novel herbicidally inactive biobased adjuvants for modern pest control. Additionally, the QSAR results revealed that the obtained compounds are potentially less toxic than their fully synthetic analogs. In effect, hexadecyltrimethylammonium and *N*-hexadecylpyridinium bromides were included in the more harmful ‘acute I’ toxicity class according to the GHS criteria. The performed QSAR studies have provided valuable ecotoxicological data demonstrating that nicotinamide substituted with the shortest (butyl) chain can be considered non-toxic to aquatic life. However, it should also be stressed that some methods used in prediction (especially those for cationic surfactants) still require further development and revision. Nevertheless, the use of currently available QSAR algorithms are perceived by the scientific community as an enormous step forward, which opens new perspectives for research on the toxicity and application of various synthetic and biobased QASs. It is undoubtedly important to continue the research on the substances of natural origin, which will lead to the discovery of more ‘green’ alternatives to currently used chemicals.

## Figures and Tables

**Figure 1 plants-12-00914-f001:**
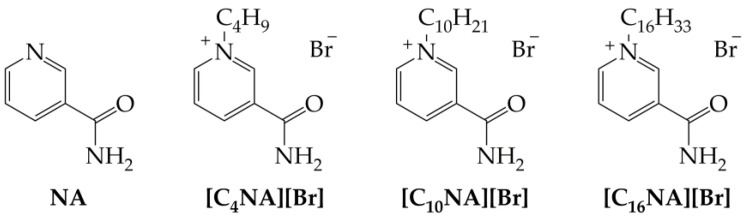
Structures of nicotinamide (NA) and its ionic derivatives used for analyses.

**Figure 2 plants-12-00914-f002:**
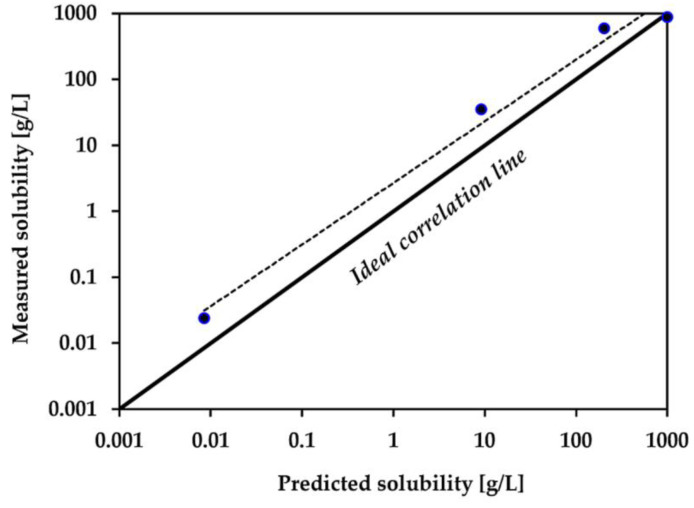
The correlation between measured and predicted values of water solubility.

**Table 1 plants-12-00914-t001:** Water solubility of the tested compounds.

Compound	Measured Solubility (g/L)	Predicted Solubility (g/L)
NA ^a^	592	204
[C_4_NA][Br]	873	1000
[C_10_NA][Br]	34.9	9.14
[C_16_NA][Br]	0.024	0.0085

^a^ The solubility of NA described in the literature is 500 or 1000 g/L [29,30].

**Table 2 plants-12-00914-t002:** Values of the logarithm of the octanol-water partition coefficient (log K_OW_) determined for the tested compounds.

Compound	Measured Log K_OW_	Predicted Log K_OW_
NA ^a^	−0.66	−0.45
[C_4_NA][Br]	−1.41	−3.10
[C_10_NA][Br]	−0.18	−0.15
[C_16_NA][Br]	1.70	2.80

^a^ Log K_OW_ of NA reported in the literature amounts to −0.37 [31].

**Table 3 plants-12-00914-t003:** Assessment of the hazards of NA and its derivatives to aquatic life.

			NA	[C_4_NA][Br]	[C_10_NA][Br]	[C_16_NA][Br]
Organism ^a^	Duration	Endpoint	Concentration [mg/L]			
Fish	96 h	LC_50_	3900	34,000	54	1.90
Daphnid	48 h	LC_50_	5700	56,000	5.9	1.60
Green Algae	96 h	EC_50_	180	1200	--- ^b^	---
*L. minor* ^c^	72 h	EC_50_	>1000	>1000	100–1000	10–24
Potential to bioaccumulation ^d^			No	No	No	No
GHS short-term toxicity category ^e^			None	None	Acute II	Acute II

Green: Above 100 [mg/L], yellow: 10–100 [mg/L], orange: 1–10 [mg/L], ^a^: The toxicity toward three representative aquatic species was predicted using suitable QSAR models, ^b^: There is no suitable QSAR model for the assessment of the toxicity of cationic surfactants toward algae, ^c^: Results determined experimentally, ^d^: Assessed using log K_OW_ and BCF, ^e^: detailed requirements for each category are described in [26].

**Table 4 plants-12-00914-t004:** Susceptibility to biodegradation estimated by the BIOWIN activated sludge models.

Compound	NA	[C_4_NA][Br]	[C_10_NA][Br]	[C_16_NA][Br]
BIOWIN 1	0.7450	0.9428	0.9027	0.8626
BIOWIN 2	0.9111	0.9794	0.9352	0.8136
BIOWIN 3	2.6609	2.8706	2.6846	2.4986
BIOWIN 4	3.8582	3.9483	3.8269	3.7055
BIOWIN 5	0.3843	0.2936	0.3140	0.3345
BIOWIN 6	0.3606	0.1540	0.1478	0.1417
BIOWIN 7	0.5276	−0.4328	−0.2769	−0.1210

**Table 5 plants-12-00914-t005:** Assessment of hazards to the aquatic environment [34,35,36].

			[C_16_TMA][Br]	[C_16_PIR][Br]	[C_16_NA][Br]
Organism	Duration	Endpoint	Concentration [mg/L]		
Fish	96 h	LC_50_	0.1–0.28	36.5; 43.5	1.90 ^a^
Daphnid	48 h	LC_50_	0.026	0.012	1.60 ^a^
Green Algae	96 h	EC_50_	0.00411	11	---
*L. minor*	72 h	EC_50_	---	---	10–24
Potential to bioaccumulation			No	No	No
GHS short-term toxicity category			Acute I	Acute I	Acute II
GHS long-term toxicity category			Chronic I	Chronic I	---

Green: Above 100 [mg/L], yellow: 10–100 [mg/L], orange: 1–10 [mg/L], red: <1 [mg/L], ^a^: predicted.

**Table 6 plants-12-00914-t006:** Summary of the phytotoxic effect demonstrated by selected compounds.

	Phytotoxicity		
Structure of Compound	Monocotyledonous Plant	Dicotyledonous Plant	Conclusions
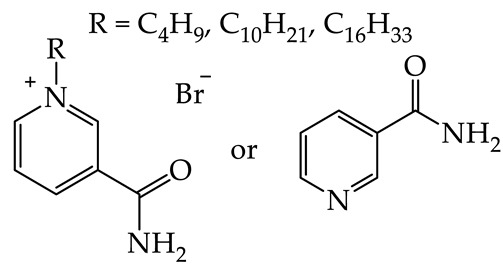	Sorghum	White mustard	Non-phytotoxic towards mono- and dicotyledonous plants.
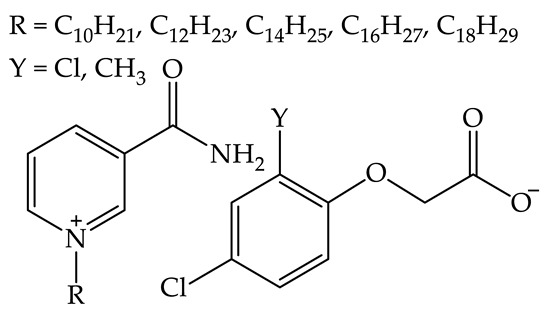	No data available	Cornflower Oilseed rape	Enhanced herbicidal effect toward dicotyledonous plants compared to original herbicides (2,4-D, MCPA). Utilized cations most likely play the role of non-phytotoxic adjuvants [21].
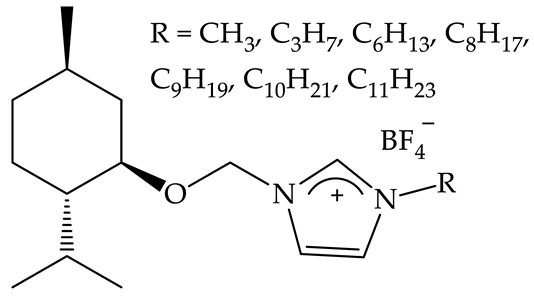	Spring barley	Common radish	The toxicity toward mono- and dicotyledonous plants decreases 6-fold with increase in alkyl chain length from C_1_ to C_11_ [17].
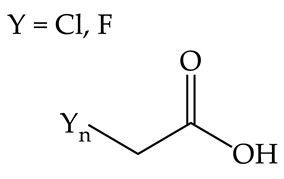	Spring barley	Common radish	Toxicity of the tested solutions increases with the number of halo-substituents and is equally high for mono- and dicotyledonous plants [39].
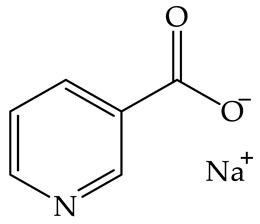	No data available	Tomato	Vitamin B_3_ in a form of sodium nicotinate applied to tomato plants can reduce the bacterial wilt of plants, while not causing phytotoxic effects [40].

## Data Availability

The data that support the findings in the present study are available from the corresponding author upon request.

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
