# Peer review of "Comprehensive Ecotoxicity Studies on Quaternary Ammonium Salts Synthesized from Vitamin B3 Supported by QSAR Calculations"

_plants, 2023, doi:10.3390/plants12040914_

Round 1
Reviewer 1 Report
The manuscript deals with the preparation on three nicotinamide quaternary ammonium salts bearing side chains of different length and the study of their phytotoxicity on two representative plants. Also, the ecotoxicological profile was investigated by means of QSAR.
The work is of interest for the readers of the journal. Therefore, I recommend it for publication provided that the following remarks are addressed.
Line 38: “proper functioning of cell metabolism functioning” the last “functioning” should be delated
Line 46: “In consequence,” should be changed. The increasing number of studies unveiled the biological properties of nicotinamide derivatives. However, they would have possessed these properties anyway; there is not a direct consequence.
Throughout the manuscript, the authors refer to as fully synthetic structures for the tetrametylammonium and pyridinium salts. This suggests that the salts obtained from nicotinamide have instead at least a natural part. However, as stated by the authors, the available nicotinamide is also “man-made” via the Lonza process. Hence, this aspect is confusing and should be amended. Indeed, the salts studied in this work are fully synthetic as well.
Spell check is needed: ie line 11 plans instead of plants, line 328 derivaieves instead of derivatives
Reviewer 2 Report
The manuscript entitled “Comprehensive Ecotoxicity Studies on Quaternary Ammonium Salts Synthesized from Vitamin B3 Supported by QSAR Calculations” investigates the impact of ionic forms of nicotinamide on the germination capacity as well as on the development of roots and stems of the tested plants. The authors claimed that, independently of the length of the alkyl chain or plant species, all QASs were established as non-phytotoxic. QSAR simulations, performed using the EPI Suite™ program pack, allowed determining products’ potential toxicity toward fish, green algae, and daphnids along with the susceptibility to biodegradation.
The below revisions are recommended:
1. First of all, the authors should mention that Vit B3 is niacin which is Nicotinic acid (Pyridine-3-carboxylic acid). It has 2 other forms, Niacinamide (nicotinamide) and Inositol Hexa Nicotinate (IHN), which have different effects on the human body from niacin although, in general, the body handles nicotinamide better than niacin since it is more similar to NAD, which results in greater bioavailability.
2. Please insert a suitable hypothesis.
3. Table 2: The measured and calculated octanol-water partition coefficient (log KOW) determination is not convincing. I recommend taking an average of three runs. When the values are small, how many times higher or lower is more important than the difference between the measured and calculated values.
4. I recommend determining the concentration of the hazard compounds (quaternary ammonium salts, QASs) by HPLC-MS. This experiment will produce more quantitative results. Then, insert the actual chromatogram in the text and indicate all the compounds as discussed in the manuscript.
5. The manuscript requires thorough revision of spelling and grammar. For example, Line# 11; plans should plants; in Line# 114; logaritm should be logarithm.
6. Uniformity (font and size) should be maintained throughout the manuscript including the schemes and figures.
Reviewer 3 Report
I would like to give some comments and recommendations to this paper:
Introduction
Lines 49-59: The text “Nicotinamide contains a pyridine ring substituted in the meta position with an amide 49 group, creating many possibilities for its use as a reagent in chemical synthesis. The main method used is the quaternization reaction of the nitrogen atom from the pyridine ring. In 1937, Karrer et al. performed and reported for the first time a successful nicotinamide quaternization with alkyl halides [11]. Over the next years, the process itself was improved in terms of increasing the yield or shortening the reaction time, where the disso-54 lution of both reactants in xylene or dimethylformamide proved to be particularly advantageous [12]. Recently, this reaction has been improved according to the principles of green chemistry, and the environmentally friendly solvent used, propan-1-ol, enabled the procedure of isolation of the desired products to be significantly simplified [13]. Presumably, the more realistic prospect of their production on a commercial scale will significantly contribute to the increased interest of the industry in these compounds in the near future.” can be removed from the text of the paper without loses. That can allow making the paper more concise. This historical information about the synthesis of the compounds is not necessary here.
Since this article was written for the Plant magazine, and not for Organic synthesis, instead of a text about the history of the synthesis of ionic derivatives of nicotinamide (NA), it is better to add text and links to the Introduction itself, with the help of which you can familiarize the reader with the advantages and disadvantages of various methods for determining the toxicity of already synthesized substances and those test-objects that will be used in this work, motivating the choice of these objects. It is better to indicate in the purpose of the study which parameters of these objects are to be evaluated for newly synthesized substances.
In addition, it should be noted that the synthesis of new ionic derivatives of nicotinamide is not listed as one of the main goals of this work. It seems that they have already been synthesized earlier and in this work only their effects on different living objects will be investigated. I propose to define the purpose of the work more clearly. I propose to consider the possibility of using in the Introduction part of the information that is given in the Discussion section.
Line 72: The “ L. Minor” should be given as “Lemna minor” because the object appeared in the text for the first time.
Lines 173, 175, 219, 309, 384, 441, Tables 3, 5, and Supplementary Materials: The “ L. Minor” should be changed to “L. minor”.
Results
I propose to partially move paragraph 2.1 to section 4.2 "Materials and methods", and to present the structures of substances themselves in the Introduction, specifying them when formulating the purpose of the study, since they and their properties are the main objects of research in the work.
At the beginning of sections 2.2 and 2.3, it is necessary to add suggestions on the purpose for which these parameters were determined. The reader should see the logical chain of actions of the authors in their research.
I disagree with the correlation presented at Figure 2. Please, present the equation of linear correlation and the value of standard square deviation. I see that the linearity is observed up to 100 g/L, but not to 1000 g /L. Performed correlation is bad, and therefore it is necessary to represent 2 straight lines – theoretical and experimental.
I recommend giving a description of the color solutions used in Tables 3 and 5, since they become clear only after familiarization with Supplementary Materials.
Discussion
Lines 245-247: the drugs synthesized and studied in the work are doubtful as inexpensive
Lines 241-247: Since the aim of the work is indicated here ("One of the objectives of the following research was to determine whether the tested solutions of NA and its derivatives affect the early development of seeds of model terrestrial plants (white mustard and sugar sorghum) ... In recent years, much attention has been paid to the possible replacement of conventional agrochemicals with inexpensive substances of natural origin due to the potential economic and environmental benefits.", then there should be a result of comparing the data obtained with those known from the literature. Please, add such a comparison in the form of an additional Table in Discussion. It will increase the significance of the results obtained. In addition, such a Table will allow readers to summarize the text that is presented in Discussion. In this Table, please, give the structures of substances discussed in the text, specific properties by columns and references that are mentioned in this part of the text. This will improve the perception of the presented discussion.
Lines 369-371: One of the most important results of the work is the study and confirmation of biodegradation of synthesized substances ("The importance of results gathered for ionic derivatives of pyridine underlines the fact that the majority of conventional, fully synthetic QASs containing imidazolium, tetraalkylammonium, morpholinium, piperidinium and pyrrolidinium cations are mostly resistant to biodegradation"). This property distinguished the results obtained from previously known analogues, and it is necessary to reflect this fact in the title of the article, keywords and in the purpose of the study in the Introduction.
Supplementary Materials
In tables S8-S10, the Duration, days column should be unified and everything should be given in days.
In tables S8-S9, gray text is poorly visible.
In Table S10, you should remove the Endpoint column and give it as a comment under the table.
Line 302 Give the full name of the microorganism R. solanacearum.
References
More than 50% of links are older than 5 years. If possible, it is necessary to update the links in the text, replacing them with more up-to-date data, especially in the Introduction and Discussion parts.
Check the design of links 23,29, 38,39,49.
Check typos throughout the text.
Round 2
Reviewer 2 Report
The authors have revised the manuscript considering majority of my recommendations. However, I am not convinced with the method of determining the concentration of the hazard compounds (quaternary ammonium salts, QASs). In their response, the authors indicated the OECD guidelines which also talks about HPLC/MS method that I recommended (please see the attached file).

Reviewer 3 Report
The authors made a notable work with the text of manuscript. I have a minor remark: it is necessary to pay attention to the fact that on line 227 there is a reference to Tables S1 and S2, then on line 254-255 it is indicated "Tables S5-S10", and the reference to Table S3 and S4 is indicated on line 259, that is, references to Tables of Supplemental Materials are given out of order in the text.
Round 3
Reviewer 2 Report
Although HPLC/MS (TOF-SIMS) or UV-Vis/Elemental analysis would be a better option to characterize the compounds that have been discussed in the manuscript, but I respect the authors’ statement, ‘due to the lack of our experience and easy access to the apparatus, we anticipate that the creation of the appropriate HPLC methodology and its validation will require many weeks or even months, therefore, we are unable to provide these results in the reasonable amount of time.’ Based on this I recommend the manuscript for publication.